**Data Availability Statement:** All relevant data are within the manuscript and its Supporting Information files.

**Funding:** Mesa-Varona O. This work was carried out within the framework of the ARDIG project, the

## RESEARCH ARTICLE

# Phenotypical antimicrobial resistance data of clinical and non-clinical *Escherichia coli* from poultry in Germany between 2014 and 2017

**Octavio Mesa-Varona**[1]*, **Heike Kaspar**[2,3], **Mirjam Grobbel**[1], **Bernd-Alois Tenhagen**[1]

1 Department Biological Safety, German Federal Institute for Risk Assessment, Berlin, Germany,
2 Department Method Standardisation, Reference Laboratories, Resistance to Antibiotics, Berlin, Germany,
3 Unit Monitoring of Resistance to Antibiotics, Federal Office of Consumer Protection and Food Safety, Berlin, Germany

* Octavio.Mesa-Varona@bfr.bund.de

## Abstract

Antimicrobial resistance (AMR) is a global threat in humans and animals, and antimicrobial usage (AMU) has been identified as a main trigger of AMR. The purpose of this work was to compare data on AMR in clinical and non-clinical isolates of *Escherichia coli* in German broilers and turkeys between 2014 and 2017. Furthermore, we investigated AMR changes over time and the association of changes in AMU with changes in AMR. Data on clinical and non-clinical isolates together with data on therapy frequency of broilers and turkeys were collected from German monitoring systems. Logistic regression analyses were performed to assess the association between the explanatory factors (AMU, year and isolate type) and the dependent variable (AMR). In broilers, the analysis showed lower resistance proportions of clinical isolates of *E. coli* to ampicillin and colistin (ampicillin: Odds ratio (OR) and 95% confidence interval (CI) = 0.44 (0.3–0.64), $p<0.001$; colistin: OR and 95% CI = 0.75 (0.73–0.76), $p<0.001$) but higher proportions for cefotaxime (OR and 95% CI = 4.58 (1.56–15.1), $p = 0.007$). Resistance to ampicillin, gentamicin and tetracycline was less frequent in clinical isolates in turkeys (ampicillin: OR and 95% CI = 0.4 (0.29–0.53), $p<0.001$; gentamicin: OR and 95% CI = 0.5 (0.26–0.94), $p = 0.035$; tetracycline: OR and 95% CI = 0.4 (0.29–0.55), $p<0.001$). The analysis found decreasing associations of AMU with resistance to tetracycline in turkeys and to colistin in broilers. Year was associated with a decrease in resistance to colistin in broilers and to tetracycline in turkeys. Differences in resistance found in this study between clinical and non-clinical isolates might play an important role in resistance prevalence. This study indicated that further data analyses over longer time intervals are required to clarify the differences found between clinical and non-clinical isolates and to assess the long-term effects of changes in AMU on the prevalence of AMR.

## Introduction

Antimicrobial resistance (AMR) is a global threat that has increased in recent years in humans and animals. Antimicrobial usage (AMU) has been identified as a main trigger of AMR [1, 2].

European Joint Programme (EJP) on AMU and AMR in humans, food and animals. ARDIG project has received funding from the European Union's Horizon 2020 research and innovation programme under Grant Agreement No 773830. URL: https://ec.europa.eu/programmes/horizon2020/en. The funders had no role in study design, data collection and analysis, decision to publish, or preparation of the manuscript.

**Competing interests:** The authors have declared that no competing interests exist.

An increase of AMU is expected in most underdeveloped countries in the coming years [3]. Large differences in AMU [4] and AMR [5–8] shown across the countries such as Spain, Italy, Norway or Sweden evidence clearly this relationship.

Global strategies have been developed to tackle this threat such as the Global Action Plan (GAP) of the World Health Organization (WHO) [9] or the new European One Health Action Plan against AMR [10]. The "Deutsche Antibiotika-Resistenzstrategie" (DART), is the national action plan (NAP) in Germany. It was first set up in 2008 in line with the recommendations made at the European level.

As a part of NAPs, surveillance and monitoring systems are essential to gather crucial information such as prevalence, incidence, trends, resistance patterns and key drivers of resistance. The systems may help to improve the global understanding of AMR helping decision makers to take appropriate actions to minimise or even prevent the spread of AMR [11]. NAPs also promote many governmental initiatives and projects that collect valuable information addressing new prevention strategies.

In relation to the collection of AMR data, historically two types of bacterial populations have been established: (a) The population collected from animals without underlying pathologies (non-clinical data; i.e. commensals) and (b) the population from diseased animals (clinical data).

In Europe, the majority of AMR data on non-clinical isolates from livestock come from standardized monitoring systems based on Commission Implementing Decision 2013/652/EU. Data are collected by the European Food Safety Authority (EFSA) [12]. On the other hand, the VetPath monitoring system, an initiative funded by the pharmaceutical industry, collects data on clinical isolates in livestock at European level [11, 13, 14]. However, the number of isolates is limited and data is not freely available. Some European countries have additionally set up programmes collecting data on clinical isolates from animals (e.g. France, Norway, United Kingdom and Germany) [11].

In Germany, in recent decades, poultry and particularly the broiler meat sector has increased its relevance as a meat source. In 2019, poultry production reached 1,918,000 tons carcass weight, of which 1,340,000 were from broilers [15].

While Europe banned antimicrobial grow promoters in 2006 [16], antimicrobials are still widely used in the poultry sector [17–19]. They are prescribed/administered to the flocks as a therapy against diseases or during metaphylactic treatment. Antimicrobials approved for use in poultry in Germany are neomycin, spectinomycin, amoxicillin, ampicillin, benzylpenicillin, phenoxymethyl-penicillin, trimethoprim, lincomycin, tylosin, tilmicosin, tylvalosin, tiamulin, colistin, enrofloxacin, sulfadimethoxine, sulfadimidine, sulfamethoxazole, sulfaquinoxaline, doxycycline and oxytetracycline [20].

*Escherichia coli* are Gram-negative bacteria commonly found in the intestine of animals as commensal microorganisms. They are also a main threat for the poultry sector causing animal disease and considerable economic losses [20]. *Escherichia coli* may serve as a reservoir spreading resistance genes horizontally to other bacteria [21]. The emergence of AMR due to AMU can be evaluated through the monitoring of resistant *E. coli*, a widely accepted AMR indicator [8, 22–24]. The relation between AMU and AMR has been extensively described in livestock in general [25], in pigs [26–28], in cattle [28–30], and in poultry [28, 31–34]. A large number of publications evaluate the *E. coli* resistance proportions in poultry without analytically considering AMU [35–42].

In 2019, a study carried out in Estonia collected AMR data on clinical and non-clinical isolates in pigs and cattle. In this study, higher proportions of resistance were observed in clinical isolates than in non-clinical isolates on the descriptive level [43], but no statistical analysis comparing both isolate types was carried out. To our knowledge, there are no publications comparing data on AMR in clinical and non-clinical isolates from poultry.

The main objective of this work is therefore to compare data on AMR in clinical and non-clinical isolates of *E. coli* from German broilers and turkeys. It would be reasonable to expect the level of resistance to be higher in clinical isolates compared to non-clinical isolates, as diseased broilers and turkeys may carry bacteria resistant to regular antimicrobial treatments [43]. Furthermore, we investigate AMR changes over time and the association of changes in AMU with changes in AMR. We challenge two hypotheses in this manuscript: (1) The level of AMR in *E. coli* from broilers and turkeys is higher in clinical isolates than in non-clinical isolates and (2) there is a demonstrable association between changes in AMU and changes in AMR in isolates from broilers and turkeys. In order to challenge our two hypotheses, we applied univariate and multivariate logistic regression analyses comparing resistance prevalence of clinical and non-clinical isolates of *E. coli* from broilers and turkeys. Further variables also included in the analyses were: (1) year (2014 to 2017) and (2) AMU (in broilers and turkeys).

## Materials and methods

### Data collection and processing

Phenotypic resistance data on clinical and non-clinical isolates of *E. coli* were collected from two different sources from 2014 to 2017. Data on non-clinical isolates from caecal samples originated from the German Zoonosis-Monitoring programme (ZoMo) [44]. Data on clinical isolates from different sample types originated from the German Resistance Monitoring of Veterinary Pathogens (GE*RM*-VET) [45]. Data on antimicrobial susceptibility testing (AST) of clinical and non-clinical *E. coli* isolates had both been obtained by broth microdilution [44, 45].

Duplicate isolates were eliminated prior to data collection preventing bias during the analysis process. To avoid a major influence of individual isolates, data were only included in the analysis when more than 24 isolates were tested and reported per year, category (clinical/non-clinical), antimicrobial drug / antimicrobial class and animal species. The antimicrobial panel analysed included cefotaxime, ciprofloxacin, colistin, nalidixic acid, tetracycline, gentamicin and ampicillin. This panel reflected the overlap between the test panels used in the two monitoring programs for clinical and non-clinical isolates.

German AMU data for broilers and turkeys were available as total amount of active ingredient in tons and as therapy frequency (TF) [17]. Therapy frequency was selected in this manuscript as a more accurate AMU parameter expressing animal exposure in days under treatment and it was used to study the association of AMU with AMR in the animal populations. Therapy frequency had been calculated using the following formula:

$$TF = (N°At \times N°TD \times N°AI) \div N°As \tag{1}$$

Where "N°At" referred to the number of animals treated, "N°TD" to the number of treatment days, "N°AI" to the number of active antimicrobial substances and "N°As" to the average number of animals in 6 months [17]. Therapy frequency values were available per semester in the database. Data for the first semester of 2014 were not available as the obligation to record the treatments in a central database started in July 2014. Therefore, second semester data of 2014 were doubled to obtain a TF approximation of the year (Table 1). Each drug belonging to the resistance panel was compared to the TF of its antimicrobial class (Table 2).

### Antimicrobial susceptibility testing

Antimicrobial susceptibility testing was determined by broth microdilution according to CLSI standard [46]. Minimum Inhibitory Concentrations (MIC) were interpreted according to Epidemiological Cut-off values (ECOFFs) provided by EUCAST (01. September 2019) (Table 2).

**Table 1. Therapy frequency, an AMU unit applied in Germany, with antimicrobial classes of broilers and turkeys from 2014 to 2017 [17].**

| Animal species | Antimicrobial class | Therapy frequency per year | | | |
|---|---|---|---|---|---|
| | | 2014 | 2015 | 2016 | 2017 |
| **Broiler** | Aminoglycosides | 11.66 | 7.68 | 8.56 | 11.32 |
| | Cephalosporins | 0.0 | 0.0 | 0.0 | 0.0 |
| | Penicillins | 8.76 | 6.25 | 5.77 | 5.54 |
| | Polymyxins | 6.88 | 5.56 | 5.03 | 5.73 |
| | Fluoroquinolones | 3.52 | 3.68 | 2.99 | 3.17 |
| | Tetracyclines | 0.82 | 0.33 | 0.35 | 0.44 |
| **Turkey** | Aminoglycosides | 1.18 | 1.22 | 1.22 | 1.16 |
| | Cephalosporins | 0.0 | 0.0 | 0.0 | 0.0 |
| | Penicillins | 31.22 | 26.9 | 23.18 | 25.55 |
| | Polymyxins | 12.24 | 9.34 | 7.99 | 7.84 |
| | Fluoroquinolones | 6.94 | 6.24 | 5.11 | 5.08 |
| | Tetracyclines | 4.54 | 3.83 | 3.24 | 2.77 |

Isolates with a MIC up to the ECOFF, i.e. wild-type isolates (isolates without acquired/mutational resistance [8]), were considered susceptible while isolates with a MIC above the ECOFF, i.e. non-wild type isolates (isolates with acquired/mutational resistance [8]), resistant.

## Statistical analysis

Data were managed and analysed using "the Konstanz information Miner (KNIME)" tool (Version 3.7.2) and the software "R" (Version 3.4.3) using the CRAN Packages "pscl", "logistf" and "ROCR". Data were analysed by univariate and multivariate logistic regression models adopting a binomial distribution. The dependent variable was the MIC categorization by EUCAST ECOFFs (i.e. resistant ($y = 1$) or susceptible ($y = 0$)). (i) Isolate type (clinical vs. non-clinical isolates), (ii) year and (iii) TF per antimicrobial class were included as explanatory factors for each antimicrobial of the AMR panel and animal species (broilers and turkeys). A univariate analysis was performed for each animal species and antimicrobial, assessing the association of each explanatory variable with the dependent variable. In the case of the (fluoro-)quinolones, similar univariate analyses were carried out for antimicrobial class duplicating isolates (i.e. a value for each drug) and each animal species in order to assess the relationship between each explanatory factor and the outcome variable for the entire antimicrobial class.

Multivariate analysis was carried out only when more than one variable per antimicrobial/antimicrobial class and animal species in the univariate analysis showed an association to the outcome variable with a *p*-value lower than 0.1. The level of significance for the univariate and

**Table 2. Antimicrobial classes, antimicrobial agent/substance tested and epidemiological cut-offs applied to categorize antimicrobial susceptibility testing results from broth microdilution based on EUCAST (01. September 2019).**

| Antimicrobial class | Antimicrobial agent/substance tested | Epidemiological cut-off values (mg/L) defining the non-wild type |
|---|---|---|
| Penicillins | Ampicillin | >8 |
| Polymyxins | Colistin | >2 |
| (fluoro-)quinolones | Ciprofloxacin | >0.064 |
| | Nalidixic acid | >16 |
| Tetracyclines | Tetracycline | >8 |
| Aminoglycosides | Gentamicin | >2 |
| Cephalosporins | Cefotaxime | >0.25 |

multivariate analysis was a *p*-value lower than 0.05. A multivariate analysis performed in broilers for colistin showed a complete or quasi-complete separation in the logistic regression [47] providing overestimated coefficients. In this case, the outcome variable separated the combination of predictor variables. A valid method penalising the likelihood was performed to overcome this issue in this analysis [48]. For the explanatory variable "Isolate type", an odds ratio (OR) <1 indicated a lower fraction of resistance in the clinical isolates compared to non-clinical isolates. An OR >1 indicated a higher fraction of resistance in the clinical isolates compared to the non-clinical isolates. The year and the TF, in the model, were analysed as numeric variables. *p*-values were obtained by the use of Wald Chi-square test. A *p*-value of less than 0.05 was considered statistically significant.

## Results

Resistance percentages, number of resistant isolates and total number of isolates tested per year and antimicrobial in broilers and in turkeys are summarized in Tables 3 and 4 respectively. Results of the univariate and multivariate logistic regression models are shown in Tables 5 and 6.

### Broilers

A total of 185 clinical isolates and 407 non-clinical isolates were collected from broilers between 2014 and 2017. In 2014, less than 25 clinical isolates were submitted and reported and were therefore excluded from the analysis. The highest resistance proportions in non-clinical isolates (>50.0%) were observed to ampicillin (2014 and 2016), nalidixic acid (2016) and ciprofloxacin (2016). In clinical isolates, highest resistance prevalence was found to ampicillin (2017), ciprofloxacin (from 2015 to 2017) and nalidixic acid (from 2015 to 2016). High levels of resistance (30.0% < 50.0%) were also found to ampicillin (from 2015 to 2016), nalidixic acid

**Table 3. Number and proportion of resistant isolates of the tested clinical and non-clinical isolates of *Escherichia coli* reported from broilers in Germany 2014–2017.**

| Drug / drugs (class) | Type of isolate | N° of resistant / N° of tested (% resistant) | | | |
|---|---|---|---|---|---|
| | | 2014 | 2015 | 2016 | 2017 |
| **Ampicillin (penicillins)** | Clinical | 9/18 (50.0%) [a] | 23/76 (30.3%) | 16/50 (32.0%) | 23/41 (56.1%) |
| | Non-clinical | 128/230 (55.7%) | | 105/177 (59.3%) | |
| **Cefotaxime (cephalosporins)** | Clinical | 0/18 (0.0%) [a] | 3/76 (3.9%) | 4/50 (8.0%) | 2/41 (4.9%) |
| | Non-clinical | 3/230 (1.3%) | | 1/177 (1.1%) | |
| **Ciprofloxacin (fluoroquinolones)** | Clinical | 3/6 (50.0%) [a] | 50/75 (66.7%) | 31/50 (62.0%) | 22/41 (53.7%) |
| | Non-clinical | 110 (47.8) | | 106/177 (59.9%) | |
| **Ciprofloxacin and nalidixic acid ((fluoro-)quinolones) [b]** | Clinical | 11/24 (45.8%) [a] | 101/151 (66.9%) | 60/100 (60.0%) | 41/82 (50.0%) |
| | Non-clinical | 213/460 (46.3%) | | 206/354 (58.2%) | |
| **Colistin (polymyxins)** | Clinical | 1/18 (5.6%) [a] | 1/76 (1.3%) | 0/50 (0.0%) | 0/41 (0.0%) |
| | Non-clinical | 16/230 (7.0%) | | 7/177 (4.0%) | |
| **Gentamicin (aminoglycosides)** | Clinical | 1/18 (5.6%) [a] | 2/75 (2.7%) | 6/50 (12.0%) | 3/41 (7.3%) |
| | Non-clinical | 16/230 (7.0%) | | 12/177 (6.8%) | |
| **Nalidixic acid (quinolones)** | Clinical | 8/18 (44.4%) [a] | 51/76 (67.1%) | 29/50 (58.0%) | 19/41 (46.3%) |
| | Non-clinical | 103/230 (44.8%) | | 100/177 (56.5%) | |
| **Tetracycline (tetracyclines)** | Clinical | 8/18 (44.4%) [a] | 13/75 (17.3%) | 7/50 (14.0%) | 13/41 (31.7%) |
| | Non-clinical | 77/230 (33.5%) | | 49/177 (27.7%) | |

[a] not included in the analysis as less than 25 isolates were tested.

[b] considering resistance to ciprofloxacin and nalidixic acid.

**Table 4. Number and proportion of resistant isolates of the tested clinical and non-clinical isolates of *Escherichia coli* reported from turkeys in Germany 2014–2017.**

| Drug / drugs (class) | Type of isolate | N° of resistant / N° of tested (% resistant) | | | |
|---|---|---|---|---|---|
| | | 2014 | 2015 | 2016 | 2017 |
| **Ampicillin (penicillins)** | Clinical | 31/82 (37.8%) | 38/104 (36.5%) | 36/95 (37.9%) | 36/63 (57.1%) |
| | Non-clinical | 118/184 (64.1%) | | 119/188 (63.3%) | |
| **Cefotaxime (cephalosporins)** | Clinical | 0/82 (0.0%) | 0/104 (0.0%) | 2/95 (2.1%) | 0/93 (0.0%) |
| | Non-clinical | 4/184 (2.2%) | | 4/188 (2.1%) | |
| **Ciprofloxacin (fluoroquinolones)** | Clinical | 19/45 (42.2%) | 31/104 (29.8%) | 29/95 (30.5%) | 19/63 (30.2%) |
| | Non-clinical | 75/184 (40.8%) | | 61/188 (32.4%) | |
| **Ciprofloxacin and nalidixic acid ((fluoro-)quinolones) [a]** | Clinical | 59/126 (46.8%) | 55/209 (26.3%) | 52/190 (27.3%) | 32/126 (25.4%) |
| | Non-clinical | 135/368 (36.7%) | | 103/376 (27.4%) | |
| **Colistin (polymyxins)** | Clinical | 0/81 (0.0%) | 4/105 (3.8%) | 3/95 (3.2%) | 6/63 (9.5%) |
| | Non-clinical | 9/184 (4.9%) | | 17/188 (9.0%) | |
| **Gentamicin (aminoglycosides)** | Clinical | 2/80 (2.5%) | 4/104 (3.8%) | 3/95 (3.2%) | 6/63 (9.5%) |
| | Non-clinical | 19/184 (10.3%) | | 12/188 (6.4%) | |
| **Nalidixic acid (quinolones)** | Clinical | 40/81 (49.4%) | 24/105 (22.9%) | 23/95 (24.2%) | 13/63 (20.6%) |
| | Non-clinical | 60/184 (32.6%) | | 42/188 (22.3%) | |
| **Tetracycline (tetracyclines)** | Clinical | 33/80 (41.3%) | 23/104 (22.1%) | 17/95 (17.9%) | 19/63 (30.2%) |
| | Non-clinical | 103/184 (56.0%) | | 81/188 (43.1%) | |

[a] considering resistance to ciprofloxacin and nalidixic acid.

(2017) and tetracycline (2017) in clinical isolates. In non-clinical isolates, high resistance proportions (30.0% < 50.0%) were observed in 2014 to ciprofloxacin, nalidixic acid and tetracycline. Increasing resistance was encountered to ampicillin and tetracycline in clinical isolates from 2015 to 2017. Nalidixic acid and ciprofloxacin resistance prevalence in clinical isolates decreased from 2015 to 2017. Resistance proportions lower than 13.0% in clinical and non-clinical isolates were found to colistin, cefotaxime and gentamicin.

Analyses revealed that resistance to colistin, cefotaxime and ampicillin differed significantly between clinical and non-clinical isolates. Resistance to ampicillin and colistin was less frequent in clinical isolates (ampicillin: OR and 95% CI = 0.44 (0.3–0.64), $p<0.001$; colistin: OR and 95% CI = 0.75 (0.73–0.76), $p<0.001$) while cefotaxime showed higher resistance proportions in clinical isolates with an OR>1 (OR and 95% CI = 4.58 (1.56–15.1), $p = 0.007$) (Table 5). No significant differences were encountered between clinical and non-clinical isolates for the (fluoro-)quinolones. However, differences were close to significance (OR and 95% CI = 1.30 (0.98–1.73), $p = 0.064$).

An association was found between year and resistance to colistin (OR and 95% CI = 0.94 (0.93–0.94), $p<0.001$). No significant association between year and resistance to (fluoro-)quinolones was shown. However, again the association was close to being significant (OR and 95% CI = 1.12 (0.99–1.27) $p = 0.064$). The analysis showed an association between TF of broilers with colistin and AMR (OR and 95% CI = 1.07 (1.06–1.08), $p<0.001$).

## Turkeys

A total of 344 clinical isolates and 372 non-clinical isolates were collected from turkeys from 2014 to 2017. The highest resistance proportions (>50.0%) were encountered for ampicillin (2014 and 2016) and tetracycline (2014) in non-clinical isolates. In clinical isolates, highest resistance prevalence was found to ampicillin (2017). High resistance frequencies (30.0% < 50.0%) in non-clinical isolates were displayed for ciprofloxacin (2014 and 2016), nalidixic acid

**Table 5. Univariate analysis results for broilers and turkeys per antimicrobial class and per (fluoro-)quinolone drug.**

| Antimicrobial class/ drug | Animal species | Factor | p-value | OR (CI) |
|---|---|---|---|---|
| **Ampicillin** | Broiler | AM usage | 0.220 | 1.07 (0.96–1.2) |
| | | Isolate type | <0.001 | 0.44 (0.3–0.64) |
| | | Year | 0.897 | 0.99 (0.84–1.16) |
| | Turkey | AM usage | 0.792 | 1.01 (0.96–1.05) |
| | | Isolate type | <0.001 | 0.4 (0.29–0.53) |
| | | Year | 0.857 | 1.01 (0.88–1.17) |
| **Cefotaxime** | Broiler | AM usage | NA | NA |
| | | Isolate type | 0.007 | 4.58 (1.56–15.1) |
| | | Year | 0.189 | 1.42 (0.85–2.47) |
| | Turkey | AM usage | NA | NA |
| | | Isolate type | 0.095 | 0.27 (0.04–1.07) |
| | | Year | 0.999 | 1.0 (0.54–1.84) |
| **Ciprofloxacin** | Broiler | AM usage | 0.225 | 0.69 (0.38–1.25) |
| | | Isolate type | 0.05 | 1.45 (1.0–2.1) |
| | | Year | 0.028 | 1.2 (1.02–1.41) |
| | Turkey | AM usage | 0.033 | 1.23 (1.02–1.48) |
| | | Isolate type | 0.206 | 0.81 (0.59–1.12) |
| | | Year | 0.028 | 0.84 (0.72–0.98) |
| **Ciprofloxacin + nalidixic acid ((fluoro-)quinolones) [a]** | Broiler | AM usage | 0.171 | 0.75 (0.49–1.13) |
| | | Isolate type | 0.004 | 1.45 (1.12–1.88) |
| | | Year | 0.004 | 1.18 (1.05–1.32) |
| | Turkey | AM usage | <0.001 | 1.33 (1.16–1.52) |
| | | Isolate type | 0.527 | 0.93 (0.74–1.17) |
| | | Year | <0.001 | 0.79 (0.7–0.88) |
| **Colistin** | Broiler | AM usage | 0.023 | 1.82 (1.1–3.13) |
| | | Isolate type | 0.025 | 0.1 (0.01–0.48) |
| | | Year | 0.016 | 0.57 (0.35–0.88) |
| | Turkey | AM usage | 0.034 | 0.82 (0.67–0.98) |
| | | Isolate type | 0.062 | 0.52 (0.26–1.02) |
| | | Year | 0.018 | 1.48 (1.08–2.06) |
| **Gentamicin** | Broiler | AM usage | 0.661 | 1.04 (0.86–1.27) |
| | | Isolate type | 0.913 | 0.96 (0.45–1.93) |
| | | Year | 0.673 | 1.07 (0.78–1.47) |
| | Turkey | AM usage | 0.064 | $4.25 \times 10^{-6}$ ($7.82 \times 10^{-12}$–2.05) |
| | | Isolate type | 0.035 | 0.5 (0.26–0.94) |
| | | Year | 0.624 | 0.93 (0.69–1.24) |
| **Nalidixic acid** | Broiler | AM usage | 0.47 | 0.81 (0.45–1.45) |
| | | Isolate type | 0.041 | 1.46 (1.02–2.11) |
| | | Year | 0.069 | 1.16 (0.99–1.36) |
| | Turkey | AM usage | <0.001 | 1.47 (1.21–1.79) |
| | | Isolate type | 0.624 | 1.08 (0.78–1.5) |
| | | Year | <0.001 | 0.72 (0.61–0.85) |
| **Tetracycline** | Broiler | AM usage | 0.008 | 2.91 (1.32–6.46) |
| | | Isolate type | 0.007 | 0.55 (0.35–0.85) |
| | | Year | 0.127 | 0.87 (0.73–1.04) |
| | Turkey | AM usage | <0.001 | 1.68 (1.33–2.13) |
| | | Isolate type | <0.001 | 0.38 (0.27–0.51) |
| | | Year | <0.001 | 0.73 (0.63–0.85) |

[a] considering resistance to ciprofloxacin and nalidixic acid.

**Table 6. Multivariate analysis results for broilers and turkeys per antimicrobial class and per (fluoro-)quinolone drug.**

| Antimicrobial class/ drug | Animal species | Factor | p-value | OR (CI) |
|---|---|---|---|---|
| **Ciprofloxacin** | Broiler | Isolate type | 0.239 | 1.27 (0.85–1.91) |
| | | Year | 0.124 | 1.15 (0.96–1.37) |
| | Turkey | AM usage | 0.997 | 1.0 (0.47–2.08) |
| | | Year | 0.578 | 0.84 (0.45–1.53) |
| **Ciprofloxacin + nalidixic acid ((fluoro-)quinolones) [a]** | Broiler | Isolate type | 0.064 | 1.30 (0.98–1.73) |
| | | Year | 0.064 | 1.12 (0.99–1.27) |
| | Turkey | AM usage | 0.920 | 1.03 (0.59–1.78) |
| | | Year | 0.349 | 0.8 (0.5–1.26) |
| **Colistin** | Broiler [b] | AM usage | <0.001 | 1.07 (1.06–1.08) |
| | | Isolate type | <0.001 | 0.75 (0.73–0.76) |
| | | Year | <0.001 | 0.94 (0.93–0.94) |
| | Turkey | AM usage | 0.502 | 1.22 (0.68–2.18) |
| | | Isolate type | 0.016 | 0.39 (0.17–0.81) |
| | | Year | 0.129 | 2.31 (0.76–6.9) |
| **Gentamicin** | Turkey | AM usage | 0.052 | $1.2 \times 10^{-6}$ ($9.88 \times 10^{-13}$–1.02) |
| | | Isolate type | 0.028 | 0.49 (0.25–0.91) |
| **Nalidixic acid** | Broiler | Isolate type | 0.150 | 1.34 (0.9–2.0) |
| | | Year | 0.278 | 1.1 (0.93–1.31) |
| | Turkey | AM usage | 0.901 | 1.05 (0.44–2.4) |
| | | Year | 0.429 | 0.75 (0.36–1.48) |
| **Tetracycline** | Broiler | AM usage | 0.144 | 1.97 (0.8–4.92) |
| | | Isolate type | 0.101 | 0.66 (0.4–1.08) |
| | Turkey | AM usage | 0.005 | 97.92 (3.66–2502.81) |
| | | Isolate type | <0.001 | 0.4 (0.29–0.55) |
| | | Year | 0.011 | 13.84 (1.76–104.98) |

[a] considering resistance to ciprofloxacin and nalidixic acid.

[b] application of a different statistical method to overcome the perfect and quasi-perfect separation phenomenon in logistic regression.

(2014) and tetracycline (2016). In clinical isolates, high resistance proportions were encountered for ampicillin (from 2014 to 2016), ciprofloxacin (2014, 2016 and 2017), nalidixic acid (2014) and tetracycline (2014 and 2017).

Resistance to ampicillin, colistin, gentamicin and tetracycline was less frequent in clinical isolates (ampicillin: OR and 95% CI = 0.4 (0.29–0.53), $p < 0.001$; colistin: OR and 95% CI = 0.39 (0.17–0.81), p = 0.016; gentamicin: OR and 95% CI = 0.49 (0.25–0.91), $p = 0.028$; tetracycline: OR and 95% CI = 0.4 (0.29–0.55), $p < 0.001$) (Tables 5 and 6). No significant differences were encountered between clinical and non-clinical isolates for cefotaxime. However, differences were close to significance (OR and 95% CI = 0.27 (0.04–1.07), $p = 0.095$).

Analysis showed a significant association between TF with tetracyclines and resistance to tetracycline (OR and 95% CI = 97.92 (3.66–2502.81), $p = 0.005$) and between year and resistance to tetracycline (OR and 95% CI = 13.84 (1.76–104.98), $p = 0.011$). An association between TF with aminoglycosides and resistance to gentamicin was close to being significant (OR and 95% CI = $1.2 \times 10^{-6}$ ($9.88 \times 10^{-13}$–1.02) $p = 0.052$).

## Discussion

The main objective of this work was to compare *E. coli* AMR in clinical and non-clinical isolates from German broilers and turkeys. Furthermore, we wanted to investigate other potential

factors that may be associated with AMR in the isolates. Our hypotheses were: (1) The level of AMR is higher in clinical isolates than in non-clinical isolates and (2) there is an association between changes in AMU and changes in AMR in isolates. In order to challenge our hypotheses, we applied univariate and multivariate logistic regression analyses assessing the OR of resistance in *E. coli* from German broilers and turkeys to an antimicrobial panel (cefotaxime, ciprofloxacin, colistin, nalidixic acid, tetracycline, gentamicin and ampicillin) with the explanatory variable "isolate type"(clinical vs. non-clinical isolates). Further variables were included in the analyses: (1) year (from 2014 to 2017) and (2) TF of broilers and turkeys with antimicrobials.

The relationship between AMR and AMU has been described in livestock [25, 49, 50]. During the last years, Germany has reduced antimicrobial consumption in food-producing animals considerably [51]. In 2014 the sales figure for antimicrobials were 149.3 mg/Population Correction Unit (PCU)), while in 2017 this figure was reduced to 89 mg/PCU [4]. This reduction in the antimicrobial sales was also reflected in the TF data of broilers and turkeys [17]. Likewise, in Germany, the level of AMR in commensal *E. coli* from livestock was effectively reduced [5, 6]. Usage data collected at farm or veterinary level are required to better address the AMR assessment in livestock [51].

Therapy frequency and resistance data of isolates from broilers and turkeys in Germany were evaluated on a national level as an association of farm level was not possible with the available data. The minimum number of isolates per year and origin was defined to 25 isolates. EFSA set up a minimum of 10 isolates in their reporting system acknowledging that this number may be too low [25]. We increased the minimum number of isolates to address these concerns and ensure the reliability of the results.

To our knowledge, Germany is the only country that provided analogous public data available on national AMU per drug class and *E. coli* AMR in non-clinical isolates from both animal species (i.e. broilers and turkeys). Discussion of broilers and turkeys results is addressed below drug by drug.

Ampicillin resistance proportions from broilers remained stable in non-clinical isolates between 2014 and 2016. The percentage of ampicillin resistance in *E. coli* from broilers in Germany was similar to the EU average in non-clinical isolates in 2014 (55.7% vs. 58.6%) and in 2016 (59.3% vs. 58.0%) [5, 6, 28]. Ampicillin resistance percentage in clinical isolates was higher in 2017 than in years before. However, in the model, the year did not show a significant association with resistance to ampicillin in isolates. The model showed a higher probability of resistance in non-clinical isolates to ampicillin in broilers.

In Germany, TF of broilers with penicillins dropped sharply between 2014 and 2017. However, resistance prevalence in clinical and non-clinical *E. coli* isolates did not decrease. Similarly in France, no association between the use of penicillins and resistance to ampicillin was encountered as sales figures showed an abrupt reduction of penicillin sales from 2014 to 2017 in poultry [19], while ampicillin resistance proportions from broilers in non-clinical isolates did not change between 2014 (55.8%) and 2016 (55.9%) [5, 6]. However, data published from other European countries showed an association between the use of penicillins and resistance to ampicillin in isolates from healthy broilers [28]. In the Netherlands, resistance to ampicillin in non-clinical *E. coli* isolates from broilers decreased between 2014 (62.1%) and 2016 (47.0%) [5, 6] being in line with a reduction of penicillins use in the same time interval [52, 53]. In Denmark, ampicillin resistance in non-clinical isolates from pigs did not change from 2014 to 2017. In line with that, the use of penicillins did not change [54]. Longer periods with low TF with penicillins are likely to be required to obtain a reduction of ampicillin resistance in isolates from broilers in Germany.

Ampicillin resistance proportions in isolates from turkeys did not change significantly from 2014 to 2016. The percentage of ampicillin resistance in isolates from healthy turkeys in Germany was similar to the EU in 2014 (64.1% vs. 69.0%) and in 2016 (63.3% vs. 64.6%) [5, 6]. Similar to broilers, resistance prevalence to ampicillin from turkeys increased in 2017 in clinical isolates. However, the year as variable did not reveal any significant association with the resistance to ampicillin in isolates. The statistical analysis in turkeys provided a significantly higher probability of resistance in non-clinical isolates to ampicillin.

Therapy frequency with penicillins in turkeys decreased substantially from 2014 to 2016, but increased in 2017. Ampicillin resistance prevalence did not decrease either in clinical nor non-clinical isolates of turkeys. In France, no association was found between the reduction of sales figures in poultry from 2014 to 2017 [19] and ampicillin resistance in non-clinical isolates from turkeys in 2014 (64.3%) and 2016 (67.0%) [5, 6]. In Sweden, antimicrobials are not frequently used for bacterial disease treatments in poultry [55]. This is in line with comparatively low resistance proportions to ampicillin in 2014 (25.4%) and in 2016 (8.2%) [5, 6]. We did not find simultaneous data on the use of penicillins and on resistance in isolates to ampicillin from turkeys in other countries to help us discuss and clarify these results.

Similar to broilers, there is probably a need to keep low TF for longer periods in order to achieve a decrease of ampicillin resistance in isolates in Germany.

Resistance percentages of clinical and non-clinical isolates to ampicillin in *E. coli* tended to be higher in turkeys than in broilers. In line with that, TF of turkeys with penicillins was also higher (4 to 5 times). These differences in TF would be expected to exert significant differences in the prevalence of ampicillin resistance between broilers and turkeys. An explanation might be that a TF about six is still high enough to sustain these resistance levels.

Colistin resistance in clinical and non-clinical isolates from broilers did not change significantly over time. Higher resistance proportions in non-clinical isolates were found in Germany than in the EU average in 2014 (7.0% vs. 0.9%) and in 2016 (4.0% vs. 1.9%) [5, 6]. The model found higher resistance proportions in non-clinical isolates and associations of year and TF with resistance to colistin in isolates.

Therapy frequency of broilers with polymyxins decreased from 2014 to 2016 but increased in 2017. Colistin resistance showed a tendency to decrease in clinical (2015: 1.3%; 2016: 0.0%; 2017: 0.0%) and in non-clinical isolates (2014: 7.0%; 2016: 4.0%) although the difference was not significant. In the Netherlands, colistin resistance in isolates of *E. coli* from broilers was not observed in 2014 and 2016 (0.0%) [5, 6]. In line with that, the use of colistin was consistently very low from 2014 to 2017 [53]. In Sweden, colistin is not used in poultry and no colistin resistance was observed [5, 6]. Longer periods with low TF with polymyxins might be likely to be required to assess a major decrease of colistin resistance in non-clinical isolates.

Clinical and non-clinical isolates in turkeys showed both an increasing resistance frequency not being significant. Similar resistance proportions in non-clinical isolates were found in Germany and in the EU average in 2014 (4.9% vs. 7.4%) and in 2016 (9.0% vs. 6.1%) [5, 6]. The analysis found higher resistance odds to colistin in non-clinical isolates.

Therapy frequency with polymyxins of turkeys decreased from 2014 to 2017 while resistance to colistin in clinical and non-clinical isolates tended to increase over time. Apparently, the TF decrease with polymyxins did not reduce the prevalence of colistin resistance in isolates from turkeys. In Sweden colistin resistance was null in 2014 and 2016 [5, 6] being in line with the non-use of colistin in poultry. We did not find analogous data on the use of polymyxins and on resistance in isolates to colistin in turkeys from other countries to help us discuss and clarify these results. Similar to broilers, longer periods with low TF with polymyxins are likely required to observe a decrease of colistin resistance in isolates. Low resistance percentages might remain even after the use of colistin has ceased as it is the case with chloramphenicol

(banned in 1994 in Europe) [56]. While florfenicol, a drug from the phenicol family, may be used to treat poultry, no preparations containing the active substance are authorised for poultry in Germany [57]. Therapy frequency with polymyxins was 1.5 to 2 times higher in turkeys than in broilers. That was in line with the higher resistance proportions in isolates from turkeys.

The scientific community is concerned about colistin, an effective antimicrobial against multi-drug resistant gram-negative bacteria, because of the mobile colistin resistance (mcr) determinants discovered in isolates from humans and animals. Different *mcr*-genes have frequently been found in *E. coli* isolates from animals and food in Germany that were phenotypically resistant to colistin [58]. This together with the higher consumption of this drug in German livestock than in most other EU countries [4] may explain the tendency towards a higher prevalence of colistin resistance in isolates from poultry in Germany than in the rest of the EU. In Germany, *mcr*-1 occurs mainly in non-clinical isolates from poultry production while proportions in cattle and pig isolates are significantly lower [58]. This is not in line with reports from Asian countries where *mcr-1* is also widespread in pigs and cattle. This might reflect different AMU patterns between countries [58].

Gentamicin resistance in clinical and non-clinical isolates from broilers did not change significantly over time. Resistance proportions to gentamicin in isolates from healthy broilers tended to be lower in Germany than in the EU in 2014 (7.0% vs. 11.6%) and in 2016 (6.8% vs. 8.9%) [5, 6].

Therapy frequency with aminoglycosides of broilers decreased sharply between 2014 and 2015 but increased again from 2015 to 2017. This is not in line with gentamicin resistance percentages in clinical and non-clinical isolates across the years. However, other European figures showed opposite results [28]. In France, sales figures of aminoglycosides for poultry tended to increase from 2014 to 2017 and an increasing tendency of resistance to gentamicin in non-clinical isolates between 2014 (1.4%) and 2016 (3.2%) was observed [5, 6]. In the Netherlands, the use of aminoglycosides in broilers decreased from 2014 to 2016 [53]. In line with that, resistance to gentamicin in non-clinical isolates from broilers tended to decrease between 2014 (6.4%) and 2016 (4.3%) [5, 6]. Long periods with low TF of aminoglycosides might be likely required to cause a decrease in proportion of resistant isolates to gentamicin in Germany.

In turkeys, gentamicin resistance percentages in non-clinical isolates did not vary significantly between 2014 and 2016 and were similar to EU levels (2014: 10.3% vs. 10.0%, 2016: 6.4% vs. 6.2%) [5, 6]. The statistical analysis in turkeys provided a significantly higher probability of resistance data on non-clinical isolates to gentamicin. Resistance to gentamicin in clinical isolates did not change significantly either from 2014 to 2016, but tended to increase between 2016 (3.2%) and 2017 (9.5%).

The model found a non-significant association between small changes in TF and resistant isolate percentage. We considered this relationship close to significance an artefact.

TF with aminoglycosides of turkeys remained stable from 2014 to 2017 (1.18; 1.22; 1.22; 1.16). This was in line with gentamicin resistance proportions in clinical and non-clinical isolates across the years. In Sweden, aminoglycoside use was particularly low in livestock [4]. In line with that, no gentamicin resistance was observed for turkeys in 2014 and 2016 (0.0%) [5, 6].

Therapy frequency of broilers with aminoglycosides was 7 to 11 times higher than TF of turkeys. However, gentamicin resistance proportions of broilers and turkeys were similar in clinical and non-clinical isolates. We did not find analogous data on the use of aminoglycosides and on resistance to gentamicin in isolates from turkeys from other countries to help us discuss and clarify these results.

Gentamicin itself is not approved for use in poultry in Germany but other antimicrobials from the same family (e.g. neomycin or spectinomycin) are. Similar to gentamicin, neomycin and spectinomycin inhibit the synthesis of proteins by binding to the 30s ribosomal sub-unit causing a misreading of the DNA of *E. coli*. Dissemination of AMR genes addressing this mechanism could explain resistance proportions to gentamicin in isolates of *E. coli* from poultry [59]. Further studies are required (a) to clarify why TF differences with aminoglycosides between broilers and turkeys did not affect significantly the resistance proportions in Germany and (b) to determine whether the spread of AMR genes addressing the latter action mechanism of aminoglycosides may explain gentamicin resistance in isolates from poultry.

Third generation cephalosporins are not licensed for use in poultry in the EU and therefore differences found between cefotaxime resistance data on clinical and non-clinical isolates cannot be attributed to the use of cephalosporins. In line with not using cephalosporins in poultry, cefotaxime resistance proportions in non-clinical isolates from turkeys and broilers were very low. Resistance prevalence to cefotaxime in non-clinical isolates from broilers in Germany tended to be lower than the EU average in 2014 (1.3% vs. 5.1%) and in 2016 (1.1% vs. 4.0%), but similar for turkeys (2014: 2.2% vs. 2.3%; 2016: 2.1% vs. 2.7%) [5, 6]. Cefotaxime resistance proportions in clinical isolates from broilers remained stable from 2015 to 2017 (3.9%; 8.0%; 4.9%). Resistance to cefotaxime was more likely in clinical isolates (Table 5). Cefotaxime resistance in clinical isolates from turkeys was rare, tended to be less frequent than in non-clinical isolates (OR and 95% CI = 0.27 (0.04–1.07), $p$ = 0.095) and did not change significantly from 2014 to 2017 (0.0%; 0.0%; 2.1%; 0.0%).

The ban or non-licensing of antimicrobials in food producing animals limits resistance prevalence but low resistance percentages remain. In line with that, low proportions of fluoroquinolone resistance in isolates from livestock are shown in United States [20] and in Australia [60] after the cessation of the use of fluoroquinolones.

Resistance to ciprofloxacin and nalidixic acid in non-clinical isolates from broilers was high and increased significantly between 2014 (47.8%; 44.8%) and 2016 (59.9%; 56.5%). In contrast, in the EU resistance proportions of *E. coli* isolates from broilers to ciprofloxacin and nalidixic acid did not change between 2014 (65.7%; 64.0%) and 2016 (62.6%; 59.8%) [5, 6]. In Germany, resistance proportions to these antimicrobials were lower than in the EU average in 2014. In clinical isolates from broilers, resistance percentages to ciprofloxacin and nalidixic acid decreased between 2015 (66.7%; 67.1%) and 2017 (53.7%; 46.3%) showing an opposite trend in resistance frequency to data on non-clinical isolates. A similar contrary resistance trend for fluoroquinolones as an entire family (i.e. nalidixic acid + ciprofloxacin) was found in clinical (2015: 66.9%; 2016: 60.0%; 2017: 50.0%) and non-clinical isolates (2014: 46.3%; 2016: 58.2%).

No significant differences were encountered between clinical and non-clinical isolates for (fluoro-) quinolones in general. However, differences were approaching significance (OR and 95% CI = 1.30 (0.98–1.73), $p$ = 0.064).

Therapy frequency with fluoroquinolones in broilers decreased non-linearly from 2014 to 2017. Minor TF increases were encountered between 2014 and 2015 and between 2016 and 2017. This is in contrast to increasing resistance proportions to ciprofloxacin and nalidixic acid in non-clinical isolates between 2014 and 2016 but in line with decreasing resistance proportions to ciprofloxacin and nalidixic acid in clinical isolates. In France, sales figures of fluoroquinolones for poultry decreased between 2014 and 2016 [19] and resistance in non-clinical isolates to ciprofloxacin and nalidixic acid in broilers decreased accordingly between 2014 (44.2%; 42.0%) and 2016 (35.6%; 34.0%) [5, 6]. Likewise, in Netherlands, the use of fluoroquinolones in broilers decreased from 2014 to 2017 [53] and resistance to ciprofloxacin and nalidixic acid in non-clinical *E. coli* isolates from broilers tended to decrease accordingly between 2014 (47.6%; 44.6%) and 2016 (41.0%; 39.3%) [5, 6]. However, some other countries have

reported no associations between the use of fluoroquinolones and resistance to nalidixic acid in broilers [28]. Additionally, some farms without using any fluoroquinolone showed a substantial resistance prevalence to these drugs suggesting that fluoroquinolone resistance *E. coli* may be transferred onto farms via replacement [61]. Biosecurity seems to be an important influencing factor on fluoroquinolone *E. coli* resistance [61]. Longer periods with linear decreasing TF with fluoroquinolones including biosecurity level variables might be required to show a clear TF effect on resistance to ciprofloxacin and nalidixic acid in isolates in Germany. Further studies considering farm management (such as conventional vs. organic production and farms showing different biosecurity levels), molecular typing and genomic data variables are required to clarify differences in results between clinical and non-clinical isolates.

A close to significant association in broilers was found between year and resistance to (fluoro-) quinolones in isolates (OR and 95% CI = 1.12 (0.99–1.27) *p* = 0.064).

In turkeys, resistance of ciprofloxacin, nalidixic acid and the tested (fluoro-)quinolones in total tended to decrease over time. Resistance proportions to ciprofloxacin and nalidixic acid in non-clinical isolates were lower in Germany (2014: 40.8%; 32.6% and 2016: 32.4%; 22.3%) than the EU average (2014: 50.3%; 43.5% and 2016: 46.3%; 37.2%) [5, 6]. In clinical isolates, a major decrease was found to ciprofloxacin and nalidixic acid between 2014 and 2015 while levels remained stable from 2015 to 2017.

The TF with fluoroquinolones in turkeys decreased particularly from 2014 to 2016 and did not change between 2016 and 2017. This is in line with the decreasing tendency of fluoroquinolone resistance in clinical and non-clinical isolates in turkeys. Fluoroquinolones are not used in Sweden to treat poultry. In line with that, resistance proportions to ciprofloxacin and nalidixic acid were very low in 2014 (11.2%; 11.2%) and 2016 (5.7%; 6.3%) [5, 6].

Therapy frequency of turkeys with fluoroquinolones was 1.7 to 2 times higher than in broilers, while resistance in isolates tended to be higher in broilers. We did not find analogous data on the use of fluoroquinolones and resistance in isolates to gentamicin in turkeys from other countries.

The use of fluoroquinolones, highest priority critically important antimicrobials for humans, in mass medication in food producing animals is a public health concern [62]. Fluoroquinolone resistance proportions in isolates from poultry are lower in the United States, where the use of these antimicrobials is not allowed in livestock, in comparison to other large poultry producers where these drugs are approved [20].

Tetracycline resistance in non-clinical isolates from broilers tended to decrease between 2014 and 2016. Resistance proportions to tetracycline in non-clinical isolates from broilers were lower in Germany than the EU average in 2014 (33.5% vs. 50.1%) and in 2016 (27.7% vs 47.1%) [5, 6]. Resistance percentages to tetracycline in clinical isolates from broilers did not change between 2015 (17.3%) and 2016 (14.0%) but increased significantly in 2017 (31.7%).

Therapy frequency of broilers with tetracyclines decreased substantially between 2014 and 2015 and increased slightly from 2015 to 2017. This was in line with numerically decreasing resistance in non-clinical isolates [5, 6] and also with increasing resistance percentages in clinical isolates between 2016 and 2017.

In turkeys, tetracycline resistance proportions in non-clinical isolates decreased between 2014 and 2016. They were lower than in the EU in 2014 (56.0%% vs. 70.9%) and in 2016 (43.1% vs 64.8%) [5, 6]. Resistance prevalence to tetracycline in clinical isolates also decreased from 2014 (41.3%) to 2016 (17.9%), but increased again in 2017 (30.2%). The statistical analysis showed a higher probability of resistance in non-clinical isolates to tetracycline in turkeys.

Therapy frequency of turkeys with tetracyclines decreased continuously from 2014 to 2017. This is in line with the decreasing resistance in non-clinical isolates, but not with the

increasing resistance in clinical isolates between 2016 and 2017. The model identified significant associations of TF and year with resistance to tetracycline.

Therapy frequency of turkeys with tetracyclines was 5 to 10 times higher than in broilers. In line with that, resistance prevalence in turkeys was also higher, which supports the association of resistance to tetracycline in *E. coli* to use of tetracycline.

Tetracyclines are substances commonly used for the treatment of food producing animals representing around 28.0% of all sold veterinary antimicrobials in 2014 and around 26.0% in 2017 in Germany [4]. This is in line with the high resistance rates for tetracyclines that may be caused by continuous high use of the substances in the animal population [28, 32].

The differences shown between clinical and non-clinical isolates underline the necessity to have clinical and non-clinical data collection systems in place. At European level, data on non-clinical isolates are collected by the EFSA surveillance system while data on clinical isolates are not yet being collected by European institutions on a routine basis. Only the VetPath monitoring system, financed by the pharmaceutical industry, collects data on clinical isolates in livestock in Europe [11, 13, 14].

In clinical isolates, we observed an increase in resistance from 2016 to 2017 for ampicillin and tetracycline in broilers and for ciprofloxacin, nalidixic acid, ampicillin, gentamicin, tetracycline and colistin in turkeys. This might be because *E. coli* strains carrying the respective resistance genes were introduced in the animal population from other sources. In case this phenomenon in clinical isolates keeps increasing in the following years, it could diminish the differences encountered between clinical and non-clinical isolates for these substances in our study. Further explanatory variables (e.g. molecular typing or genomic data) are required to clarify this phenomenon but were not available in our study.

The sampling frames from data on clinical and non-clinical isolates differ being able to contribute to the differences encountered in this work. Data on non-clinical and clinical isolates compared in this work differed respectively in the following aspects: (a) Mandatory (non-clinical) vs. voluntary (clinical isolates) data collection basis, (b) isolate collection at the slaughterhouse vs. during the lifetime or at time of death or during post mortem, (c) isolate collection at a fixed age vs. different ages, (d) caecal samples vs. diverse sample origins and (e) data representative for the animal population in the country vs. data representative for the samples examined in the laboratories contributing to the system. The pathogenicity of the isolates tested was not determined in this study. While it can be assumed that many of the clinical isolates were avian pathogenic *E. coli* because they were isolated from diseased animals, we did not investigate these isolates beyond their phenotypic resistance to the antimicrobials. Vice versa, the commensal E. coli isolates were from healthy animals, but this obviously does not assure that they might not be pathogenic under specific circumstances. We, therefore, chose for the terminology of clinical and non-clinical isolates rather than pathogenic or non-pathogenic isolates.

There was a significant reduction in antimicrobial sales to veterinarians in Germany and likewise in TF from 2014 to 2017 in broilers and turkeys [17]. We found associations between the year (from 2014 to 2017) and resistance to colistin in broilers and to tetracycline in turkeys. However, a significant association between TF and resistance was only found for tetracycline in turkeys and for colistin in broilers (Tables 5 and 6). This suggests that other factors not considered in this study may have had a major influence on the resistance proportions. One of those might be colonization of chicks after hatching with bacteria from the hatchery environment or carry over from previous fattening flocks in the housing environment [63, 64].

We have observed partly different trends in resistance in clinical and non-clinical isolates with an identical TF. Specific *E. coli* strains could dominate in the clinical isolates due to their pathogenicity but not in the randomly selected commensals providing a plausible explanation to the differences in results reported in this work.

Further studies with longer time ranges are required (1) to clarify the differences found between clinical and non-clinical isolates and (2) to assess the long-term effects of changes in AMU and in AMR.

## Conclusions

- In line with our hypothesis, resistance to cefotaxime was more frequent in clinical than in non-clinical isolates in broilers. In contrast, a higher probability of resistance in non-clinical isolates was encountered for ampicillin and colistin in broilers and for ampicillin, colistin, gentamicin, and tetracycline in turkeys. This suggests that other factors not considered in the manuscript, such as animal age at time of sample collection in clinical isolates, genetic data or sample type may have an effect on resistance prevalence.

- Due to the differences of trends and proportions shown in this study between clinical and non-clinical isolates, this work suggests that it is not enough to analyse data on either of the two to show a proper resistance proportion to a drug per an animal type within a country. Data on clinical isolates and non-clinical isolates should both be considered.

- Although the relationship between AMU and AMR is generally well documented, in our study the association of AMU of a drug class with AMR to a specific drug from this class was only significant for colistin in broilers and tetracycline in turkeys. This could suggest that is not enough to address AMR by reducing AMU indicating that as many influencing AMR factors as possible should be taken into consideration.

- Resistance rates to ampicillin and fluoroquinolones were among the highest in all populations. Resistance to tetracycline was highest in turkeys, but not in broilers in line with differences in AMU.

- The effect of the year was only found significant for resistance proportions to colistin for broilers and to tetracycline for turkeys. A decreasing association was only observed to colistin for broilers. It could suggest that longer periods with continuous low TF are required to demonstrate a resistance decrease in prevalence. However, as pointed out above, AMU reduction alone might not be enough in some cases to achieve a decrease in AMR.

## Supporting information

**S1 Data. Phenotypical antimicrobial resistance data of clinical and non-clinical *Escherichia coli* from broilers and turkeys in Germany between 2014 and 2017.**
(XLSX)

## Acknowledgments

We gratefully acknowledge the work of all people involved in carrying out the monitoring programs for AMR and AMU in Germany. Without them, this kind of work could not be performed.

We thank Madelaine Norström for her suggestions to the manuscript.

## Author Contributions

**Conceptualization:** Bernd-Alois Tenhagen.

**Data curation:** Octavio Mesa-Varona.

**Formal analysis:** Octavio Mesa-Varona.

**Methodology:** Octavio Mesa-Varona.

**Project administration:** Bernd-Alois Tenhagen.

**Resources:** Heike Kaspar, Mirjam Grobbel.

**Software:** Octavio Mesa-Varona.

**Supervision:** Bernd-Alois Tenhagen.

**Writing – original draft:** Octavio Mesa-Varona.

**Writing – review & editing:** Heike Kaspar, Mirjam Grobbel, Bernd-Alois Tenhagen.

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
