## [Decision Letter · Decision Letter 0]

26 Oct 2020

PONE-D-20-28854

Phenotypical antimicrobial resistance data of clinical and non-clinical *Escherichia coli* from poultry in Germany between 2014 and 2017.

PLOS ONE

Dear Dr. Mesa-Varona,

Thank you for submitting your manuscript to PLOS ONE. After careful consideration, we feel that it has merit but does not fully meet PLOS ONE’s publication criteria as it currently stands. Therefore, we invite you to submit a revised version of the manuscript that addresses the points raised during the review process.

A number of questions have been raised in methodology, presentation of results and discussion.

We look forward to receiving your revised manuscript.

Kind regards,

Iddya Karunasagar

Academic Editor

PLOS ONE

Journal Requirements:

2. Please include captions for your Supporting Information files at the end of your manuscript, and update any in-text citations to match accordingly. Please see our Supporting Information guidelines for more information: http://journals.plos.org/plosone/s/supporting-information

Additional Editor Comments:

Two reviewers have commented on the manuscript and have pointed out the need for improvement in number of aspects. Please address all the reviewer comments point by point.

Reviewers' comments:

Reviewer's Responses to Questions

**Comments to the Author**

1. Is the manuscript technically sound, and do the data support the conclusions?

Reviewer #1: No

Reviewer #2: Yes

2. Has the statistical analysis been performed appropriately and rigorously? 

Reviewer #1: No

Reviewer #2: Yes

3. Have the authors made all data underlying the findings in their manuscript fully available?

Reviewer #1: Yes

Reviewer #2: Yes

4. Is the manuscript presented in an intelligible fashion and written in standard English?

Reviewer #1: Yes

Reviewer #2: Yes

5. Review Comments to the Author

Reviewer #1: I have enjoyed reading your manuscript entitled " Phenotypical antimicrobial resistance data of clinical and non-clinical Escherichia coli from poultry in Germany between 2014 and 2017. "

Please find my suggestions and recommendations bellow.

1) The explanatory variable year is not included properly in your models. It should be included as a categorical variable, choosing year 2014 as a referent and comparing all the other years (2015, 2016, and 2017) to year 2014. You have included year in your model as a continuous variable , which is not the case.

2) I think when you discuss your study and present associations between AMR and AMU you should specify each time that you tested associations between resistance to a particular drug (i.e. Ampicillin) and use of Ampicillin in broiler and turkey flocks. It is confusing using the AMU term all the time, because it implies AMU in general, which is not your study's objective.

3) In addition you have tested associations between AMR and AMU for certain antimicrobials (i.e. cephalosporins) that are not allowed to be used in poultry production according to your paper (Lines 73-76). Consequently, it is not appropriate to compare clinical and non clinical E.coli isolates for cephalosporins because clinical cases might receive cephalosporins as a treatment.

You could consider testing associations among resistance to cephalosporins and β-lactam use as β-lactams includes penicillin derivatives, cephalosporins , monobactams, carbapenems and carbacephems. Some of these antimicrobials are allowed to be used in poultry.

4) You need to define that the majority of clinical E. coli isolates are Avian Pathogenic Escherichia coli and I understand that you did not test for virulence genes, which define an APEC isolate, but we can imply that those isolates are APEC.

Please consider the following research article:

Varga C, Brash ML, Slavic D, Boerlin P, Ouckama R, Weis A, Petrik M, Philippe C, Barham M, Guerin MT. Evaluating Virulence-Associated Genes and Antimicrobial Resistance of Avian Pathogenic Escherichia coli Isolates from Broiler and Broiler Breeder Chickens in Ontario, Canada. Avian Dis. 2018 Sep;62(3):291-299. doi: 10.1637/11834-032818-Reg.1. PMID: 30339507.

5) You should discuss and consider in your analysis co-selection for resistance and multidrug resistance. Several studies demonstrated this issue.

6) You should consider comparing AMR patterns between chicken and turkey isolates, and identify which poultry type has increased odds of resistance.

Please read the research article bellow that used this approach:

Varga C, Guerin MT, Brash ML, Slavic D, Boerlin P, Susta L. Antimicrobial resistance in fecal Escherichia coli and Salmonella enterica isolates: a two-year prospective study of small poultry flocks in Ontario, Canada. BMC Vet Res. 2019 Dec 21;15(1):464. doi: 10.1186/s12917-019-2187-z. PMID: 31864357; PMCID: PMC6925488.

7) I would suggest to change the "data source" explanatory variable to "pathogenic E.coli ad non-pathogenic E.coli", because data source implies the origin of the data but it is not specific enough.

8) You should consider using most recent and more specific (related to poultry and E.coli and AMU) at the Introduction section. Please consider the following articles:

Roth N, Käsbohrer A, Mayrhofer S, Zitz U, Hofacre C, Domig KJ. The application of antibiotics in broiler production and the resulting antibiotic resistance in Escherichia coli: A global overview. Poult Sci. 2019 Apr 1;98(4):1791-1804. doi: 10.3382/ps/pey539. PMID: 30544256; PMCID: PMC6414035.

Luiken REC, Van Gompel L, Munk P, Sarrazin S, Joosten P, Dorado-García A, Borup Hansen R, Knudsen BE, Bossers A, Wagenaar JA, Aarestrup FM, Dewulf J, Mevius DJ, Heederik DJJ, Smit LAM, Schmitt H; EFFORT consortium. Associations between antimicrobial use and the faecal resistome on broiler farms from nine European countries. J Antimicrob Chemother. 2019 Sep 1;74(9):2596-2604. doi: 10.1093/jac/dkz235. PMID: 31199864; PMCID: PMC6916135.

9) Please describe the choice of including data if at least 25 isolates were tested. Perhaps to have enough variability and power in your analysis?

10) Line 150-151. What are the random and systematic components in your model? Did you account for sample level clustering?

11) Line 160. Why did you include in the multivariable model only variables significant at 0.05 . Usually a more relaxed p value is used (i.e.0.1)

12. Line 244. The associations in a logistic regression models are signified by an odds ratio. You did not assessed resistance proportions. It is more appropriate to describe it as "assessing the odds of resistance"

Thank you for considering my suggestions!

Reviewer #2: The publication is written in a very clear and understandable way. The data is well structured. However, I have some suggestions for the improvement of the manuscript.

Abstract: Suggestion to add the meaning of all abbreviations, like odds ratio (OR).

181-182 In Table 5 – suggestion to place p-value in one line.

The text from 262-264 “To our knowledge, Germany is the only country that provided analogous public data available on national AMU and E. coli AMR in non-clinical isolates from both animal species (i.e. broilers and turkeys)” This is not correct. There are other counties (UK, France) in the EU that provide public data on AMR in clinical and non-clinical E. coli isolates. AR data for E. coli isolated from broilers in the United Kingdom are available from two distinct AR-monitoring programs: the EU monitoring and the clinical monitoring programs (Veterinary Medicines Directorate, 2015). The EU monitoring program isolated E. coli from healthy broilers across the United Kingdom. The clinical monitoring program is passive monitoring. France participates in the EU monitoring of AR in animals and detects resistance rates of non-clinical E. coli from broilers since 2005. The French Agency for Veterinary Medicinal Products (ANSES-ANMV) reports on national antibiotic resistance monitoring in France. The ANSES-ANMV provides reports on the French surveillance network for antibiotic resistance in pathogenic bacteria of animal origin (RESAPATH). The RESAPATH presents the results of the monitoring of AR in E. coli from diseased hens and broilers that are treated by veterinarians as part of their regular clinical services.

510. The main driver of resistance in poultry remains antimicrobial use in the higher level of production pyramid- primary breeder, breeder and hatchery level. The use of antimicrobial considered in this level should be added to the reason why the TF in broilers/turkeys is not the only influencing factor on the prevalence of AMR. Therefore, antimicrobial use in breeders and in the hatchery may be the reason for higher resistance rates in the non-clinical isolates. It can be mentioned in the conclusion as well as in the abstract.

6. PLOS authors have the option to publish the peer review history of their article (what does this mean?). If published, this will include your full peer review and any attached files.

Reviewer #1: No

Reviewer #2: **Yes: **Nataliya Roth

---

## [Author Response · Author response to Decision Letter 0]

5 Nov 2020

PONE-D-20-28854

Phenotypical antimicrobial resistance data of clinical and non-clinical Escherichia coli from poultry in Germany between 2014 and 2017.

PLOS ONE

Dear Dr. Mesa-Varona,

Thank you for submitting your manuscript to PLOS ONE. After careful consideration, we feel that it has merit but does not fully meet PLOS ONE’s publication criteria as it currently stands. Therefore, we invite you to submit a revised version of the manuscript that addresses the points raised during the review process.

A number of questions have been raised in methodology, presentation of results and discussion.

We look forward to receiving your revised manuscript.

Kind regards,

Iddya Karunasagar

Academic Editor

PLOS ONE

Journal Requirements:

 Answer:Done

2. Please include captions for your Supporting Information files at the end of your manuscript, and update any in-text citations to match accordingly. Please see our Supporting Information guidelines for more information: http://journals.plos.org/plosone/s/supporting-information

Answer:Done

Additional Editor Comments:

Two reviewers have commented on the manuscript and have pointed out the need for improvement in number of aspects. Please address all the reviewer comments point by point.

Reviewers' comments:

Reviewer's Responses to Questions

Comments to the Author

1. Is the manuscript technically sound, and do the data support the conclusions?

Reviewer #1: No

Reviewer #2: Yes

2. Has the statistical analysis been performed appropriately and rigorously? 

Reviewer #1: No

Reviewer #2: Yes

3. Have the authors made all data underlying the findings in their manuscript fully available?

Reviewer #1: Yes

Reviewer #2: Yes

4. Is the manuscript presented in an intelligible fashion and written in standard English?

Reviewer #1: Yes

Reviewer #2: Yes

5. Review Comments to the Author

Reviewer #1: I have enjoyed reading your manuscript entitled " Phenotypical antimicrobial resistance data of clinical and non-clinical Escherichia coli from poultry in Germany between 2014 and 2017. "

Please find my suggestions and recommendations bellow.

1) The explanatory variable year is not included properly in your models. It should be included as a categorical variable, choosing year 2014 as a referent and comparing all the other years (2015, 2016, and 2017) to year 2014. You have included year in your model as a continuous variable , which is not the case.

Answer: We agree that the “year” variable may be included in the statistical analysis as a categorical variable to compare one year against the others. In our case, we applied the “year” as a continuous variable in order to look for a resistance trend (positive or negative) across the years. Therefore, an OR >1 indicates an increasing resistance trend and an OR <1 a decreasing resistance trend across the years. 

2) I think when you discuss your study and present associations between AMR and AMU you should specify each time that you tested associations between resistance to a particular drug (i.e. Ampicillin) and use of Ampicillin in broiler and turkey flocks. It is confusing using the AMU term all the time, because it implies AMU in general, which is not your study's objective. 

Answer: Done

3) In addition you have tested associations between AMR and AMU for certain antimicrobials (i.e. cephalosporins) that are not allowed to be used in poultry production according to your paper (Lines 73-76). Consequently, it is not appropriate to compare clinical and non clinical E.coli isolates for cephalosporins because clinical cases might receive cephalosporins as a treatment.

Answer: We agree we cannot explain the differences found to cefotaxime between clinical and non-clinical isolates with the use of cephalosporins. We have included it in the discussion. Actually, we expected to find similar resistance proportions for cefotaxime in clinical and non-clinical isolates in broilers and in turkeys, as cephalosporins are not applied. However, we found higher resistance proportions in clinical than in non-clinical isolates in broilers. We thought it interesting to report these differences despite our unability to explain them. The reasons behind the differences might be clarified in the future including other variables such as genetic studies in the isolates. 

You could consider testing associations among resistance to cephalosporins and β-lactam use as β-lactams includes penicillin derivatives, cephalosporins , monobactams, carbapenems and carbacephems. Some of these antimicrobials are allowed to be used in poultry.

Answer: We agree that more detailed analyses of the association between AMU and AMR beyond the direct association studied here might add further insights. However, co-resistance is not something specific for penicillins and cephalosporins. Hence it would have changed the whole approach to the data had we included all assumed co-resistance effects available in the literature. We did not perform it as we focussed on the association between AMU per class and resistance to a drug belonging to the latter class. 

4) You need to define that the majority of clinical E. coli isolates are Avian Pathogenic Escherichia coli and I understand that you did not test for virulence genes, which define an APEC isolate, but we can imply that those isolates are APEC.

Answer: We agree that some of E. coli isolates collected from diseased animals may surely be APEC. However, we do not know whether E. coli collected was the cause of the death / illness of the animal. Therefore, we cannot assume that. However, in the discussion we address this point briefly to explain why we write on clinical and non-clinical vs. pathogenic and non pathogenic isolates.

Please consider the following research article:

Varga C, Brash ML, Slavic D, Boerlin P, Ouckama R, Weis A, Petrik M, Philippe C, Barham M, Guerin MT. Evaluating Virulence-Associated Genes and Antimicrobial Resistance of Avian Pathogenic Escherichia coli Isolates from Broiler and Broiler Breeder Chickens in Ontario, Canada. Avian Dis. 2018 Sep;62(3):291-299. doi: 10.1637/11834-032818-Reg.1. PMID: 30339507.

Answer: Done

5) You should discuss and consider in your analysis co-selection for resistance and multidrug resistance. Several studies demonstrated this issue.

Answer: We agree that several studies demonstrate co-selection for resistance and multidrug resistance. However, we did not perform this approach as we have no molecular data.

6) You should consider comparing AMR patterns between chicken and turkey isolates, and identify which poultry type has increased odds of resistance.

Answer: We agree this is an interesting question that may be clarified in further studies. We did not aim to compare resistance data between broilers and turkeys but within animal populations. We aim 1)to identify higher resistance proportions in clinical or non-clinical isolates per animal species and 2) to evidence a relationship between AMU and AMR per animal species. We mention this issue once with respect to resistance to tetracycline (lines 481-483), but did not aim for a thorough comparison between the species as this had added a multitude of other factors (differences in lifespan, husbandry etc.) that we had no data on.

Please read the research article bellow that used this approach:

Varga C, Guerin MT, Brash ML, Slavic D, Boerlin P, Susta L. Antimicrobial resistance in fecal Escherichia coli and Salmonella enterica isolates: a two-year prospective study of small poultry flocks in Ontario, Canada. BMC Vet Res. 2019 Dec 21;15(1):464. doi: 10.1186/s12917-019-2187-z. PMID: 31864357; PMCID: PMC6925488.

7) I would suggest to change the "data source" explanatory variable to "pathogenic E.coli ad non-pathogenic E.coli", because data source implies the origin of the data but it is not specific enough.

Answer: As we indicated in the comment 4, we cannot assume that strains from clinical E. coli isolates are pathogenic. We replaced “data source” by “isolate type”.

8) You should consider using most recent and more specific (related to poultry and E.coli and AMU) at the Introduction section. Please consider the following articles:

Roth N, Käsbohrer A, Mayrhofer S, Zitz U, Hofacre C, Domig KJ. The application of antibiotics in broiler production and the resulting antibiotic resistance in Escherichia coli: A global overview. Poult Sci. 2019 Apr 1;98(4):1791-1804. doi: 10.3382/ps/pey539. PMID: 30544256; PMCID: PMC6414035.

Answer: This reference was already cited in the introduction (Line 77).

Luiken REC, Van Gompel L, Munk P, Sarrazin S, Joosten P, Dorado-García A, Borup Hansen R, Knudsen BE, Bossers A, Wagenaar JA, Aarestrup FM, Dewulf J, Mevius DJ, Heederik DJJ, Smit LAM, Schmitt H; EFFORT consortium. Associations between antimicrobial use and the faecal resistome on broiler farms from nine European countries. J Antimicrob Chemother. 2019 Sep 1;74(9):2596-2604. doi: 10.1093/jac/dkz235. PMID: 31199864; PMCID: PMC6916135.

Answer: Done (Line 85)

9) Please describe the choice of including data if at least 25 isolates were tested. Perhaps to have enough variability and power in your analysis?

Answer: We indicated in the manuscript (line 263-264) that we increased the minimum number of isolates to 25 based on the minimum of 10 isolates set up by the EFSA in the JIACRA reports to ensure the reliability of the results. The choice of the number as always an arbitrary choice between loosing data by increasing the threshold and reducing the effect of specific isolates. Changing this number to 40 would have had no effect as the next smallest population would have been clinical isolates from broilers in 2017 with 41 isolates. 

10) Line 150-151. What are the random and systematic components in your model? Did you account for sample level clustering?

Answer: This sentence has been removed

11) Line 160. Why did you include in the multivariable model only variables significant at 0.05 . Usually a more relaxed p value is used (i.e.0.1). 

Answer: Thank you for this suggestion. We changed the analysis but only minor changes were observed. Those are shown in track changes mode.

12. Line 244. The associations in a logistic regression models are signified by an odds ratio. You did not assessed resistance proportions. It is more appropriate to describe it as "assessing the odds of resistance"

Answer: Done

Thank you for considering my suggestions!

Reviewer #2: The publication is written in a very clear and understandable way. The data is well structured. 

Answer: Thank you.

However, I have some suggestions for the improvement of the manuscript.

Abstract: Suggestion to add the meaning of all abbreviations, like odds ratio (OR).

Answer: Done

181-182 In Table 5 – suggestion to place p-value in one line. 

Answer: Done

The text from 262-264 “To our knowledge, Germany is the only country that provided analogous public data available on national AMU and E. coli AMR in non-clinical isolates from both animal species (i.e. broilers and turkeys)” This is not correct. There are other counties (UK, France) in the EU that provide public data on AMR in clinical and non-clinical E. coli isolates. AR data for E. coli isolated from broilers in the United Kingdom are available from two distinct AR-monitoring programs: the EU monitoring and the clinical monitoring programs (Veterinary Medicines Directorate, 2015). The EU monitoring program isolated E. coli from healthy broilers across the United Kingdom. The clinical monitoring program is passive monitoring. France participates in the EU monitoring of AR in animals and detects resistance rates of non-clinical E. coli from broilers since 2005. The French Agency for Veterinary Medicinal Products (ANSES-ANMV) reports on national antibiotic resistance monitoring in France. The ANSES-ANMV provides reports on the French surveillance network for antibiotic resistance in pathogenic bacteria of animal origin (RESAPATH). The RESAPATH presents the results of the monitoring of AR in E. coli from diseased hens and broilers that are treated by veterinarians as part of their regular clinical services.

Answer: Thank you for bringing this up. The UKVARSS (UK) and the “Sales survey of veterinary medicinal products containing antimicrobials in France” do not report on AMU per individual animal species (i.e. broiler or turkey) and drug class (This is a general issue). In the case of the Netherlands, we found data on AMU for broilers and turkeys per drug class but only data on AMR per drug for broilers. This is probably because turkey production in the Netherlands is not large. We did not find analogous public data available on national AMU per drug class and E. coli AMR in non-clinical isolates from individual animal species (i.e. broilers or turkeys).

Regarding the data on clinical isolates, these data are published for only those countries that have this kind of system in place (these systems are voluntary). In contrast to data on non-clinical isolates, data on clinical isolates are not harmonised applying in most cases different standards based on different laboratory methods and procedures that are used to interpret data. Therefore, data on clinical and non-clinical isolates cannot be directly compared. For example, RESAPATH (France) applies the CASFM standard while the Scanning system (UK) uses the BSAC standard.

510. The main driver of resistance in poultry remains antimicrobial use in the higher level of production pyramid- primary breeder, breeder and hatchery level. The use of antimicrobial considered in this level should be added to the reason why the TF in broilers/turkeys is not the only influencing factor on the prevalence of AMR. Therefore, antimicrobial use in breeders and in the hatchery may be the reason for higher resistance rates in the non-clinical isolates. It can be mentioned in the conclusion as well as in the abstract.

Answer: We agree that breeder and hatchery level may be important factors influencing AMR. However, AMU data from the breeding level were not available in Germany rendering the level of this influence speculative. Moreover, this is true for the primary colonization of all broiler and turkey chicks. The primary aim was not to assess the level of resistance in the populations as that has been published in the reports that the data originate from, but its association with use in the populations and with respect to differences between clinical and non clinical data. We now mention this in the discussion

6. PLOS authors have the option to publish the peer review history of their article (what does this mean?). If published, this will include your full peer review and any attached files.

Do you want your identity to be public for this peer review? For information about this choice, including consent withdrawal, please see our Privacy Policy.

Reviewer #1: No

Reviewer #2: Yes: Nataliya Roth

---

## [Decision Letter · Decision Letter 1]

30 Nov 2020

Phenotypical antimicrobial resistance data of clinical and non-clinical *Escherichia coli* from poultry in Germany between 2014 and 2017.

PONE-D-20-28854R1

Dear Dr. Mesa-Varona,

We’re pleased to inform you that your manuscript has been judged scientifically suitable for publication and will be formally accepted for publication once it meets all outstanding technical requirements.

Kind regards,

Iddya Karunasagar

Academic Editor

PLOS ONE

Additional Editor Comments (optional):

All reviewer comments have been addressed.

Reviewers' comments:

Reviewer's Responses to Questions

**Comments to the Author**

1. If the authors have adequately addressed your comments raised in a previous round of review and you feel that this manuscript is now acceptable for publication, you may indicate that here to bypass the “Comments to the Author” section, enter your conflict of interest statement in the “Confidential to Editor” section, and submit your "Accept" recommendation.

Reviewer #1: All comments have been addressed

Reviewer #2: All comments have been addressed

2. Is the manuscript technically sound, and do the data support the conclusions?

Reviewer #1: Yes

Reviewer #2: Yes

3. Has the statistical analysis been performed appropriately and rigorously? 

Reviewer #1: Yes

Reviewer #2: Yes

4. Have the authors made all data underlying the findings in their manuscript fully available?

Reviewer #1: Yes

Reviewer #2: Yes

5. Is the manuscript presented in an intelligible fashion and written in standard English?

Reviewer #1: Yes

Reviewer #2: Yes

6. Review Comments to the Author

Reviewer #1: Thank you for addressing all of my suggestions and comments.

I know that your research article will provide valuable information on antimicrobial resistance and use issues in broiler and turkey flocks that ultimately will mitigate the development of antimicrobial resistance in commensal and pathogenic bacteria of poultry.

I have only minor editing suggestions, listed bellow.

Lines 327-328 "The analysis found higher resistance proportions to colistin in non-clinical isolates." As you have used logistic regression the association is the odds ratio and not proportions, and the sentence should be changed to "The analysis found higher odds of resistance to colistin in non-clinical isolates."

Lines 510-511 "We there chose for the terminology of clinical and non-clinical isolates rather than pathogenic or non pathogenic isolates." it should be changed to " “We, therefore, chose the terminology of clinical and non-clinical isolates rather than pathogenic and nonpathogenic isolates”

Thank you!

Reviewer #2: The authors of the publication have evaluated the comments of reviewers. The manuscript can be accepted for publication.

7. PLOS authors have the option to publish the peer review history of their article (what does this mean?). If published, this will include your full peer review and any attached files.

Reviewer #1: No

Reviewer #2: No

---

## [Editor Report · Acceptance letter]

3 Dec 2020

PONE-D-20-28854R1 

Phenotypical antimicrobial resistance data of clinical and non-clinical *Escherichia coli* from poultry in Germany between 2014 and 2017. 

Dear Dr. Mesa-Varona:

I'm pleased to inform you that your manuscript has been deemed suitable for publication in PLOS ONE. Congratulations! Your manuscript is now with our production department. 

Kind regards, 

on behalf of

Dr. Iddya Karunasagar 

Academic Editor

PLOS ONE